# Dynamic Variations of 3′UTR Length Reprogram the mRNA Regulatory Landscape

**DOI:** 10.3390/biomedicines9111560

**Published:** 2021-10-28

**Authors:** Estanislao Navarro, Adrián Mallén, Miguel Hueso

**Affiliations:** 1Experimental Nephrology Lab., Institut d’Investigació Biomèdica de Bellvitge-IDIBELL, C/ Feixa Llarga s/n, L’Hospitalet de Llobregat, 08907 Barcelona, Spain; amallen@idibell.cat; 2Department of Nephrology, Hospital Universitari Bellvitge, and Institut d’Investigació Biomèdica de Bellvitge-IDIBELL, C/ Feixa Llarga s/n, L’Hospitalet de Llobregat, 08907 Barcelona, Spain

**Keywords:** alternative 3′UTRs, 3′UTR shortening, alternative splicing, alternative polyadenylation and cleavage, Alu exonization

## Abstract

This paper concerns 3′-untranslated regions (3′UTRs) of mRNAs, which are non-coding regulatory platforms that control stability, fate and the correct spatiotemporal translation of mRNAs. Many mRNAs have polymorphic 3′UTR regions. Controlling 3′UTR length and sequence facilitates the regulation of the accessibility of functional effectors (RNA binding proteins, miRNAs or other ncRNAs) to 3′UTR functional boxes and motifs and the establishment of different regulatory landscapes for mRNA function. In this context, shortening of 3′UTRs would loosen miRNA or protein-based mechanisms of mRNA degradation, while 3′UTR lengthening would strengthen accessibility to these effectors. Alterations in the mechanisms regulating 3′UTR length would result in widespread deregulation of gene expression that could eventually lead to diseases likely linked to the loss (or acquisition) of specific miRNA binding sites. Here, we will review the mechanisms that control 3′UTR length dynamics and their alterations in human disorders. We will discuss, from a mechanistic point of view centered on the molecular machineries involved, the generation of 3′UTR variability by the use of alternative polyadenylation and cleavage sites, of mutually exclusive terminal alternative exons (exon skipping) as well as by the process of exonization of Alu cassettes to generate new 3′UTRs with differential functional features.

## 1. Introduction

Almost all the eukaryotic mRNAs are similarly structured, with a central coding region flanked by two segments of non-coding sequences, the 5′ and 3′ untranslated regions (UTRs), and a 3′ polyadenylated end-tail of variable length [1,2]. The stability and correct spatiotemporal translation of mRNAs are ensured by 3′UTRs, which harbor regulatory signals. These are sequence motifs that bind to different effectors such as RNA-binding proteins, miRNAs or other ncRNAs. Recent works have evidenced that 3′UTRs are highly polymorphic in length, and that a single gene is able to express a number of different 3′UTRs that differ in length, sequence and assembly of regulatory motifs. Thus, control of 3′UTR length regulates accession of different effectors to 3′UTR functional boxes to establish differential regulatory landscapes for mRNA function [3,4]. Here, we will review recent literature on the mechanisms that control 3′UTR length dynamics and their alterations in human disorders. We will specifically center on alternative polyadenylation and cleavage (APA), alternative splicing and on the generation of new 3′UTRs by exonization of Alu cassettes.

## 2. Sequences at the 3′UTR Regulate mRNA Stability

The finding that 3′UTRs contained segments with a high degree of sequence-conservation across species [5] changed the view of mRNA-3′UTRs from simple stabilizing end-regions to complex and highly structured regulatory platforms that controlled how, when and where the genetic information was translated. From a mechanistic point of view, the functional organization of mRNA-3′UTRs is based on short-sequence, cis-regulatory elements that are recognized and bound by trans-regulatory factors, either RNA binding proteins (RBPs) or other RNAs, mainly miRNAs [4]. Usually, 3′UTR-cis-acting sequences are structured in clusters of repeated modules that are recognized on a sequence basis by RBPs with repeated RNA-binding motifs (RBMs) or by miRNAs and other ncRNAs [4,6], although some of them are able to interact with the secondary structure of the binding motif [7]. Most of these regulatory motifs have a critical role in the stability of mRNAs as we will show in the following sections.

### 2.1. Stability Determinants in the 3′UTRs of Eukaryotic mRNAs Recognized byRNA Binding Proteins

Initial detection of a number of messenger RNAs that were targeted for selective and rapid degradation allowed the description of the first motifs for mRNA stability, i.e., the conserved AU-rich elements (AREs) at the 3′UTR of unstable cytokine/chemokine mRNAs [8,9]. AREs, the best known of mRNA destabilizing motifs, are short sequences (50–150 bases long) that include one to several copies of the pentanucleotide AUUUA in an AU-rich context and recruit the degradation machinery to ARE-containing mRNAs, although a few destabilizing regions have been also described that lack the AUUUA box [10]. Other binding motifs have been also associated with the control of mRNA stability. Thus a pyrimidine-rich sequence [(U/C)(C/U)CCCU] motif within the 3′UTR of tyrosine hydroxylase mRNA increased its stability during hypoxia [11], while a highly conserved UC-rich region in the 3-UTR of the androgen receptor mRNA that contained 5′-C(U)(n)C and 3′-CCCUCCC motifs was shown to reduce expression of a luciferase reporter gene through binding to HuR and to poly(C)-binding proteins-1 and-2 (CP1/2) [12].

The intracellular fate of mRNAs depends on the complex interplay among degradation or stabilization-promoting RBPs, whose equilibrium determines mRNA half-life. Just as an example, stability of the *VEGF* mRNA in response to hypoxia relied on the balance among degradation-promoting ARE-binding factors, such as the ACTH-regulated zinc-finger proteins Tis11,Tis11b and Tis11d [13], and the stabilization factors HuR that bound to U-rich motifs [14] or hnRNPL that interacted with an AC-rich sequence [15]. In a similar way, tristetraprolin (TTP) was shown to destabilize *TNFα* mRNA after binding directly to the AU-rich region of its 3′UTR [16], AUF1 proteins (also known as hnRNP-D proteins) were characterized as destabilizing factors of the *α-globin* mRNA by binding to AREs [17], and KSRP was shown to promote mRNA degradation by associating to AREs through its KH domain [18]. On the contrary, hnRNPsA1, E1 and K were shown to stabilize *collagen* (*Col*) mRNAs by binding to CU-and AU-rich binding motifs [19], while the *uPA* mRNA was stabilized by the binding of hnRNPC to an AU-rich sequence “in vitro” [20] and HuR, a member of the ELAV family of RNA binding proteins, was shown to stabilize ARE-containing mRNAs by binding to its ligands SETα/β, pp32 and APRIL [21]. Lastly, *insulin* mRNA was stabilized by the binding of polypyrimidine tract-binding protein (PTB) to a pyrimidine-rich sequence in its 3′UTR [22].

### 2.2. Stability Determinants Recognized by miRNAs in the 3′UTRs of Eukaryotic mRNAs

miRNAs are small RNAs with a critical role in the regulation of gene expression. Although most miRNAs (canonical miRNAs) are transcribed by RNA polymerase II recent work has highlighted a number of non-canonical miRNAs with alternative biogenesis [23]. miRNAs target the 3′UTRs of mRNAs for degradation or translational arrest, although there are also reports on their association to coding regions especially in genes with short 3′UTRs [24]. It is estimated that the entire human miRNAome is composed by over 2300 mature miRNAs [25] that can potentially interact with over 60% of total human mRNAs [26], creating complex regulatory networks in which single miRNAs can target dozens of different mRNAs which in turn can be regulated by many different miRNAs [27].

Although miRNA binding sites (MBSs) are usually unique in an individual mRNA, 3′UTRs may include multiple binding sites for different miRNAs, or even for the same miRNA [28], that can be organized in clusters of overlapping binding sequences [29]. The determinants for miRNA binding in the 3′UTRs of mRNAs are not restricted to the miRNA sequence but may also include contextual features in the form of neighboring protein binding sites or other sequential structural frameworks that modulate miRNA binding to its cognate sites [30]. In this sense, it has been described that protein *puf-9* (a *pumilio* homolog) and miRNA *let-7* bound to closely associated neighbor sites in the 3′UTR of the *hunchback* (*hbl-1*) mRNA to cooperatively repress its translation in *C.elegans* [31], while on the contrary, binding of DND1 to U-rich regions at 3′UTRs protected target mRNAs by blocking miRNA binding sites in human cells [32]. Lastly, a GC-rich motif frequently found downstream of miRNA target sites has been shown to increase stability of miRNA–mRNA duplexes [33], while a 27-nucleotide sequence between two *let-7* binding sites at the 3′UTR of the *lin-41* mRNA in *C.elegans*, was required for the silencing effect of *let-7* on *lin-41* [34].

Another critical requirement for effective miRNA–mRNA interaction is the physical accessibility of miRNAs to their binding sites, since 3′UTRs have the potential to form highly folded structures “in vivo” that could modulate the accession of miRNAs to their sites [35]. Thus, miRNA binding would require the local unwinding and unpairing of the target site to promote the nucleation of a miRNA–mRNA seed whose elongation would form the stable duplex [36], since structured stems overlapping the 5′ seed or 3′ complementary regions interfered with miRNA binding to cognate sites [37].

## 3. Temporal/Spatial Determinants of Translation in 3′UTRs: The Case of Maternal mRNAs

Control of mRNA function by RBPs and miRNAs that target 3′UTR stability determinants is not a stationary regulatory mechanism restricted to specific tissues or developmental stages, but a very dynamic process that includes temporal or spatial (different subcellular locations) regulatory axes. A relevant model to study the mechanisms of spatio-temporal regulation of mRNA translation is the oocytic cell in its first stages of development, which are regulated by a pool of mRNAs (known as maternal mRNAs) that are loaded onto the cell cytoplasm and account for over 7000 different transcripts [38,39]. A subset of these maternal mRNAs is maintained functionally inactive in oocytes until the maternal-to-zygotic transition (MZT), when these are recruited onto polysomes and translated in the absence of transcription [40,41]. After this translational repression is released, maternal mRNAs are actively degraded and transcription takes then charge of the regulation of gene expression [42]. Translational control of maternal mRNAs is a complex issue that involves the binding of RBPs to specific 3′UTR motifs and AU-rich cytoplasmic polyadenylation elements (CPEs) [43]. In the next section, we will review with more detail these mechanisms regulating stability and function of maternal mRNAs.

### 3.1. Determinants for the Temporal Regulation of Stability and Translation of Maternal mRNAs in the Maternal-to-Zygotic Transition

Unlike the regulatory mechanisms that control stability and translational arrest of somatic mRNAs, maternal mRNAs require the activity of the 5′/3′ RNA degradation machinery on the 3′ poly-A tail. Furthermore, while most eukaryotic RNAs require nuclear 3′end polyadenylation to initiate nuclear export and translation [44], several maternal mRNAs are submitted to cytoplasmic polyadenylation to activate their translation precisely at the maternal-to-zygotic transition, among them mRNAs that encode proteins involved in transcriptional or developmental regulation, dorsoventral patterning, gastrulation or germ layer formation [45].

A few RBPs have been identified that promote stability of maternal mRNAs, among them RBPY-box binding protein2 (MSY2) [46] the insulin-like growth factor 2 mRNA binding protein3 (IGF2BP3) [47] or the cytoplasmic poly-A polymerase Wispy [48]. On the other hand, embryonic maturation leads to the phosphorylation of MSY2 and to the subsequent activation of dormant maternal mRNAs that encode components of the RNA degradation machinery [46] such as the members of the DCP1A-DCP2 decapping complex, the subunits 6l and 7 of the deadenylation machinery CCR4-NOT (CNOT6l/7), the deadenylase complex CAF1-CCR4-NOT-interacting protein CUP [49], and the poly-A specific ribonuclease subunit 2 (PAN2) [50]. Deadenylation results in mRNA destabilization, translational inactivation [51] and the clearance of maternal mRNAs by the poly-A-specific exoribonuclease (PARN) [52], miR-430 [53] and the RNA m^6^A-reader, YTHN^6^-methyladenosine RNA binding protein 2 (YTHDF2) [54]. Other factors also promote maternal mRNA activation and instability by interacting with different 3′UTR motifs, with 3′ poly-A tails or even with other RBPs, such as Smaug or members of the Musashi family (MSI). Smaug (Smg) is a major regulator of the destabilization of a significant number of maternal mRNAs at the MZT [39,55]. Smaug (also known as translational repressor of nanos [56]) is a RBP that binds to motifs at the 3′UTR of mRNAs such as *Staufen*, *TIAR*, *TIA1* and *HuR* [57] as well as of maternal mRNAs to recruit the CCR4/NOT deadenylase complex and induce mRNA degradation [58]. Interestingly, Smaug from *Drosophila* has been seen to recruit Ago1 to the 3′UTR of *nanos* in a miRNA-independent manner, likely a new mechanism of mRNA degradation [59]. Lastly, Musashi is another activator of dormant maternal mRNAs that target mRNAs through its interaction with the RBPs poly-A-binding proteins ePABP or PABPC1 [60].

### 3.2. Determinants for the Intracellular Spatial Localization of Maternal mRNAs in Oocytes

Appropriate spatial translation is ensured by restricting the intracellular distribution of mRNAs through the binding of specific RBPs, to the extent that asymmetric distribution of mRNAs is a widespread regulatory mechanism of gene expression. Genetic studies in a number of cell systems have provided some insights on the mechanisms and genes promoting and maintaining an asymmetric subcellular distribution of certain mRNAs (see [61] for a recent review).

One of the most convenient cell systems to study the asymmetric distribution of mRNAs is the oocytic cell. In the case of *Drosophila*, the body plan of the insect can be traced down to the first stages of oogenesis, in which a complex interplay of regulatory mechanisms in the form of RNPs produced by the neighbour nurse cells, ensure the loading and asymmetric intracellular distribution of maternal determinants, their precise translational regulation, and the asymmetric cell divisions needed to generate daughter cells with distinct cytoplasmic contents and developmental fates [62]. Briefly, the establishment of an antero/posterior (a/p) axis is controlled by the asymmetric loading of the *bicoid* (*bcd*) mRNA to the anterior pole, and of *nanos* (*nos*) and *oskar* (*osk*) mRNAs to the posterior pole of the oocyte in a way dependent on the microtubule net work, on molecular motors of the dynein and kinesin families [63], as well as on the presence of a conserved YUGUUYCUG box in the 3′UTRs of *bicoid* and *nanos* mRNAs [64]. Fertilization will activate translation of *bicoid* and *nanos* so that the asymmetric distribution of their protein products would generate a functional a/p axis.

## 4. Regulation of 3′UTR Length by Alternative Polyadenylation or Alternative Splicing

The sequencing revolution has evidenced the high degree of variation in the length of 3′UTRs of mRNAs, and its subsequent impact on the regulatory landscapes of mRNA function. Regulation of 3′UTR length is thus becoming an important research topic in the control of gene expression by its potential to regulate mRNA-protein or mRNA–RNA interactions, and consequently mRNA function [4,65]. In this section we will deepen on the mechanisms known to regulate 3′UTR length by alternative polyadenylation and cleavage (APA) or alternative splicing (AS), as well as their alterations in human disorders (Table 1).

### 4.1. Regulating 3′UTR Length by Alternative Polyadenylation

Pre-messenger RNAs (pre-mRNAs) are specifically cleaved and polyadenylated at precise positions of their 3′ ends in a way determined by a specific polyadenylation signal (PAS) and executed by a number of multiprotein complexes. The polyadenylation signal (AAUAAA in its canonical form) positions the cleavage/polyadenylation specificity factor complex (CPSF) close to the cleavage site, over 30 nts from the specific site at which the pre-mRNA will be cleaved and the poly(A) tail will be added by the PAP activity ([66] for a review). Cleavage precision is ensured by the presence of two U-rich, upstream (USE) and downstream (DSE) sequence elements next to the AAUAAA signal that help to distinguish functional polyadenylation sites from randomly occurring hexamers [67]. These U-rich sites are recognized by the multiprotein complexes cleavage factor I (CFI) and cleavage/polyadenylation specificity factor (CPSF) which bind to the U-rich USE and AAUAAA hexamer, respectively, and the cleavage stimulation factor CstF binding to the U-rich DSE [68].

Over 70% of human genes have more than one polyadenylation site in their 3′UTRs and 50% have three or more [69], while in mouse liver over 60% of expressed genes harbor multiple polyadenylation signals in their 3′UTRs [70]. Use of different sites makes alternative polyadenylation a widespread mechanism to regulate gene expression by generating transcript variants that are heterogeneous in length and show alternative 3′ ends [71,72] (Figure 1A). When linked to alternative splicing of terminal exons, APA originate very complex patterns of 3′UTR variability, as in the case of human *DDX3X* mRNA whose three terminal untranslated exons originate six 3′UTR splicing variants that harbor five different alternative PAS [73]. Additionally, 3′UTR-alternative mRNA isoforms may show differential functional features with regards to stability [74], translational efficiency [75], microRNA binding potential [76], and tissue specific expression [77], subcellular localization at the mRNA [78] or at the protein level [79] or interactions with the network of competing endogenous RNAs (ceRNAs) [80]. Nevertheless, other authors have proposed that alternative 3′UTR sequences would have a reduced regulatory impact compared to other mRNA regions [81].

The emerging picture on the use of alternative PASs proposes that 3′UTR shortening would increase mRNA stability by relaxing protein or miRNA-based mechanisms of mRNA degradation, while 3′UTR lengthening would strengthen accessibility to miRNAs [82]. In any case, alterations in the mechanisms regulating 3′UTR length would result in widespread deregulations of gene expression that could eventually lead to diseases, likely linked to the loss of specific miRNA binding sites (Table 1) [83,84]. While long 3′UTRs have been mostly detected in quiescent stem cells, differentiated cells, or early in development, shortened 3′UTRs have been mostly described in quickly cycling cells such as proliferative stem/progenitor cells or tumour cells [85,86,87]. Thus, shortened 3′UTRs have been reported in proliferation-related transcripts in triple negative breast cancers [88], non-small cell lung cancer [89], glioblastoma [90], esophageal carcinoma [91], multiple myeloma [92], colorectal cancer [93] cardiac hypertrophy [94], etc. Furthermore, aberrant APA dynamics has been also described in acute myeloid leukemia and mature erythroid cells [95], neurological disorders [96], hypertrophic heart [97], heart failure [98] as well as in arsenic stress [99], while neurons were shown to express 3′UTRs longer than other brain cells [100].

APA-depending shortening of 3′UTRs is caused by changes in the expression of different components of the cleavage/polyadenylation complexes, i.e., cleavage factors CFI (composed by CPSF5/CFIm25/NUDT21, CPSF6-7) and CFII (PCF11, CLP1), cleavage and polyadenylation specificity factors CPSF (CPSF1-4, WDR33, FIP1L1) or cleavage stimulation factors CSTF1-3 ([101,102] for reviews).

After transcriptomic analysis of HeLa cells in which CPSF5/CFIm25/NUDT21 had been down-regulated, Masamha et al. identified 1450 transcripts with shortened 3′UTRs because of the use of proximal polyadenylation signals (pPAS), among them known oncogenes, and characterized CPSF5/CFIm25/NUDT21 as an inhibitor of proximal polyadenylation [90,124]. Furthermore, expression of CPSF5/CFIm25/NUDT21 was reported to be down-regulated in the lungs of patients with idiopathic pulmonary fibrosis (IPF) or mice with pulmonary fibrosis [125], in HEK293 cells [126], and in low/high grade glioma cells [127], causing a global shortening of 3′UTRs. Down-regulation of CPSF5/CFIm25/NUDT21 was seen to promote shortening of the 3′UTRs of *IGF1R*, *CCND1* and *GSK3**β* mRNAs and to increase transcript stability in lung adenocarcinomas and lung squamous cell carcinomas when compared to normal controls [128]. Lastly, CPSF5/CFIm25/NUDT21 not only promoted the use of distal PAS but also increased the efficiency of miRNA-mediated gene silencing by facilitating binding to AGO2 [129].

Other polyadenylation and cleavage factors also promoted the use of proximal PAS, such as Pcf11 or Fip1 in C2C12 cells [130], CPSF6 in human hepatocellular carcinoma cells [131], CSTF2 and CPEB3 in lung adenocarcinoma cells [132], or CSTF2 in bladder carcinoma, through shortening of the *RAC1*-3′UTR [133], or in TAMR breast cancer cells by shortening and stabilizing the 3′UTR of *HuR* [134]. Lastly, SRSF7, NUDT21 and HuD also favored usage of proximal polyadenylation signals, resulting in shorter 3′UTRs [135,136,137]. 

On the contrary, other factors directly promoted the use of distal PAS, such as PabpC/N1 in C2C12 cells [130] and SRSF3 in neuronal and 293T cells [135,138], or caused 3′UTR lengthening by displacing other factors as ADAR1 [139]. Lastly, a premature polyadenylation signal created in the *CCND1* gene by a single-point mutation promoted a shortening of its 3′UTR that increased the risk to mantle cell lymphoma [140].

### 4.2. Regulating 3′UTR Length by Alternative Splicing

Splicing-based mechanisms, such as alternative or cryptic splicing of terminal untranslated exons or the integration of repetitive elements through exonization also have the potential to generate 3′UTR length variants ([123,141] and Figure 1). Nevertheless, the impact of splicing-based mechanisms of 3′UTR lengthening is lower than those based in alternative polyadenylation and cleavage, as highlighted by a bioinformatic analysis on the superfamily of odorant receptor (*OR*) genes that showed that over 80% of *OR* mRNAs were submitted to alternative polyadenylation while only a few of these used alternative splicing to generate variant 3′UTRs [142].

Regulation of alternative splicing is a very complex topic that involves multiple regulatory sites at the pre-mRNAs (splicing donor and acceptor sites, canonical, cryptic and alternative sites, splicing enhancers and silencers, etc.) that are recognized by a plethora of mRNA binding proteins, U-small nuclear RNAs and associated proteins ([143] for review). In this sense, and as a general rule, introns would be identified by the binding of U1snRNP and the U2AF65/U2AF35 complex to the splice sites [144].

Basically, a few splicing mechanisms originate 3′UTR length variability and modify 3′UTR regulatory potential, i.e., intron retention, exon skipping, incorporation of one of two mutually exclusive terminal exons of different length or activation of cryptic splice sites that modify the relative lengths of the ORF and 3′UTR ([145] and Figure 1B). In this sense, a number of splicing regulators have been implicated in the regulation of 3′UTR length. Thus, cytoplasmic polyadenylation element binding protein1 (CPEB1) was seen to mediate shortening of 3′UTRs by changing patterns of alternative splicing through repression of U2AF65 recruitment or by influencing the use of alternative polyadenylation sites [146], and splicing of the 3′UTR of *Yes1-associated transcriptional regulator* (*YAP*) mRNA was seen to be dependent on hnRNPF [147]. Furthermore, SR protein kinase SPK-1 promoted 3′UTR splicing of polarity protein *Par-5* mRNA [148], and quaking (QK), a global regulator of splicing [149], was seen to promote stability of *hnRNPA1*, a repressor of alternative splicing, by binding a conserved 3′UTR sequence [150]. Lastly, expression of splicing factors ESRP1, PTB and SF2/ASF, was significantly altered in cardiac hypertrophy, leading to removal of instability-promoting AT-rich elements from 3′UTRs [151]. As for the case of APA, these splicing-based changes of 3′UTR length were associated with changes in the binding patterns of miRNAs (Table 1).

A special case of 3′UTR-lengthening caused by alternative splicing is the retention of non-coding introns [152]. Authors reported that a significant number of transcripts included retained introns in their 3′UTRs that harbored miRNA binding sites (in 387 out of the 2864 human genes analyzed [117]), or Staufen2 (Stau2) sites (in 356 transcripts [153]). Interestingly the presence of an alternative retained intron in the 3′UTR of splicing factor *SRSF1 (ASF/SF2)* protected this isoform from NMD degradation in HCT116 colon cancer cells [154].

## 5. Regulation of 3′UTR Length by Exonization of Repeated Sequences of the Alu Family

Mammalian 3′UTRs harbor a number of mobile genetic elements from the Short/Long Interspersed Nuclear Element (SINE/LINE) families [141]. Among them, the most important is the SINE family of Alu repeats, highly successful genomic invaders (over 10^6^ Alu elements are present per diploid genome), that originate from the processing of the 7SL RNA component of the signal recognition particle (SRP) [155] which colonized the human genome through repeated cycles of retrotranscription-retrogression [156]. Alu elements can be found in clusters in intergenic or intronic regions, or embedded in transcriptional units, mostly in the 3′UTRs of mRNAs but also in their 5′UTR or coding regions [157]. The genomic localization dictates the transcriptional fate of Alu elements, with intergenic “free” Alus being transcribed by RNA Pol III from their own internal promoters while mRNA-embedded Alus are transcribed by RNA Pol II from the promoter of the transcriptional units harboring them [158]. Although most of the Alu elements are currently stable genomic fossils, a very small number of Alu elements (called “young” or “master”Alus) are still retrotransposition-competent [159], and still cause genomic alterations by insertional mutagenesis, abnormal expression, or by facilitating recombinatory events among them ([160] for review).

Alu repeats impact on mRNA function by another, poorly known, mechanism termed “exonization”, by which the splicing machinery incorporates, “de novo”, an intronic Alu element to a mature transcript with the subsequent structural modifications derived of the functional activation of alternative stop-codons and polyadenylation signals encoded in or downstream of the Alu element [160]. As for their “normal” Alu counter parts, exonized Alu elements are exclusively expressed in primates, thus adding an extra layer of complexity to the onset and development of human diseases. Interestingly, another mechanism for the generation of 3′UTR variability has been also described in which the Alu insertion harbours an acceptor splice site whose activation results in the intronization of the intermediate sequence and in the shortening of the 3′UTR [161]. Here, we will review data on the impact of Alu elements on the stability and function of mRNAs, centering on the effects derived from their exonization without considering other Alu-based mutagenic mechanisms.

### 5.1. Docked-Alu Elements and the Stability of 3′UTRs: Sequence-Dependent vs. Sequence-Independent Mechanisms

As discussed in the previous sections, 3′UTRs regulate mRNA stability by harbor ring specific stability/degradation sequence motifs that recruit sequence-specific protein or miRNAs. In this sense, 3′UTR-Alu repeats have been proposed as potential sites for specific miRNA binding [162,163], with target sites coinciding with conserved Alu sequences [164], although this is a highly controversial topic and other authors consider these Alu-dependent miRNA binding sites as neutral or non-functional [165]. In this sense, only a few 3′UTR-Alu/miRNA interactions have been confirmed, among them the targeting of *double minute 2/4* mRNAs (*Mdm2/4*) by miR-661 [166], or that of *RAD1, GTSE1, NR2C1, FKBP9 and UBE2l* by miR-15a-3p and miR302d-3p [116]. Other authors consider that these 3′UTR-Alus could actually work as miRNA sponges [115,167] in a way similar to that proposed for free Alus [168] or other ncRNAs [169].

In addition to the sequence-specific determinants of mRNA stability above described, other sequence-independent mechanisms have been reported that rely on the generation of Alu-dependent secondary structures in the 3′UTRs and are recognized by the A-to-I edition machinery [170], or by the Staufen-mediated Decay (SMD) pathway of mRNA degradation [171], a mechanism similar to Nonsense Mediated Decay (NMD) [172]. In these mechanisms, two Alu elements, embedded in the 3′UTR in an inverted orientation base-pair to form dsRNA duplexes potentially targeted by RBPs such as Staufen [171] or the RNA editing adenosine deaminase acting on RNA (ADAR) enzymes [173]. In the SMD, Staufen (Stau) proteins bind to dsRNA regions, including the 3′UTR base-paired Alu duplexes, and induce the degradation of the transcript [174,175] while, in A-to-I edited transcripts, ADAR activity deaminated adenosines to inosines in the 3′UTR-Alu elements [176], leading to the generation of transcripts lacking the repetitive elements and intermediate sequences by eliminating the duplexed Alus and the associated stem-looped sequence [177]. Furthermore, since inosines base-pair with cytidines, ADAR-mediated mRNA editing in 3′UTR-Alu elements could also have an impact on the binding potential of miRNAs and RBPs to 3′UTRs [178], and a recent study has detected a high number of edited sites mapping to potential miRNA target binding sites in the 3′UTRs of mRNAs in human lung tissue [179]. Lastly, A-to-I edition is a highly dynamic process and novel works have showed alterations of the editome in lung adenocarcinomas [180], leukemias [181], thyroid cancer [182], hepatocellular carcinomas [183], well as in the context of Alu exonization-dependent evolution of cancer genomes [184].

### 5.2. Exonized Alus Impact on the 3′UTRome by Activating Alternative Stop Codons, Polyadenylation Signals or Splicing Sites

Alu repeats modify the 3′UTRome through the double process of exonization and subsequent neo-functionalization of the exonized sequence [158]. Alu-repeated elements contain a number of potential 5′/3′ splice sites that facilitate their incorporation to 3′UTRs by alternative splicing [185], although these are suboptimal variants of the canonical splicing donor/acceptor signals and suggest that expression of the Alu-including variants would not be constitutive, but optional [186]. The process of Alu-exonization is relatively frequent and some authors have estimated that over 5% of alternative exons in the human genome could derive from Alu elements [187], with over 300 of these (corresponding to 243 genes) leading to the formation of new 3′ terminal variants [188]. At the mechanistic level, exonized Alus generate gene variants with alternative 3′ ends through the activation of premature stop codons [189] or through the activation of downstream cryptic polyadenylation sites (PAS) [190] that can truncate or elongate a mature transcript depending on the location of the Alu-derived sequence (Figure 1C). In this sense, over 10,000 Alu elements are harbored in 3′UTRs of human protein-coding genes, of which more than one hundred have the ability to reprogram the 3′UTR length by providing functional polyadenylation and cleavage sites (PAS) to their transcripts [191]. These 3′UTR-Alus are mainly found in the forward sense orientation and show hot-spots of PAS-cleavage signals [188], mainly in their A-rich linker region between the two Alu arms, or in the short polyA tails of the Alu elements which can mutate to canonical AAUAAA polyadenylation signals [192].

The molecular mechanisms promoting the inclusion of exonized Alus into mature transcripts are complex and poorly studied [193]. A first mechanism relies in the competition for cryptic and functional splice sites between the exonization-promoting splicing factor U2AF65 and the suppressive hnRNP-C1/C2 that displaces the former from splice sites and represses exonization. Deregulation of this process, e.g., by mutations in hnRNP-C binding sites, would cause the aberrant activation of cryptic splice sites and result in Alu exonization [194]. Another mechanism for exonization involves the ADAR-depending reaction of A-to-I edition that converts adenosines (A) to inosines (I) which subsequently base-pair to cytosines (C) [195]. This process occurs mostly in the non-coding regions of mRNAs, and over 90% of all A-to-I editing events have been traced to 3′UTR-docked Alu elements [176]. A-to-I editing of Alu repeats originated new splice donor (AU edited to IU=GU) and acceptor (AA edited to AI=AG) sites that contribute to the exonization of intronic Alus by alternative splicing [170]. ADAR acts on tandem Alu repeats with reverse orientation that facilitate duplexing (termed IRAlus [196]), in a process that is highly dependent on the distance separating the two reversed Alu elements which must be smaller than 800bp [197]. Remarkably, while duplexed Alu elements in 3′UTRs have been shown to be substrates of the viral sensor response of the innate immune response (TLRs, RIG-I and MDA5, PK-R, NLRP3 inflammasome [198]), ADAR-1 A-to-I edited duplexed Alus were characterized as suppressors of such a response [199,200]. Lastly, Alu exonization has been shown to be altered in a number of human diseases as shown in Table 2.

## 6. Conclusions and Future Trends

3′UTRs were originally considered as having only a protective role on mRNA stability, but massive sequencing of cDNAs and new RNA.seq approaches have unveiled the diversity of the UTR landscape. Now, it is widely accepted that 3′UTRs are dynamic regions whose lengths, and sometimes their entire exonic composition, are highly regulated in normal cells and altered in human disorders so that their arrangement of regulatory motifs and their networks of regulative interactions can change. In this work we have reviewed the main mechanisms modifying the 3′UTR landscape, from alternative polyadenylation and cleavage to the less frequent alternative splicing of 3′-terminal exons or the rare exonization of repetitive elements of the Alu family, mechanisms that alone or in combination increase 3′UTR variability.

This research field is dependent on the generation of sequencing data and the associated requirements, interpretation, curation, annotation and analysis of sequence data. Current development of affordable, high-throughput sequencing technologies has had a positive impact on the detection and characterization of functional 3′UTR variants, their patterns of regulatory motifs and the establishment of alternative co-regulatory networks with other mRNAs, miRNAs or ncRNAs (lincRNAs, ceRNAs, etc.). Nevertheless, new technical developments are required to fully exploit the impact of 3′UTR variability and its alterations on the mechanisms regulating gene expression in human disorders, and especially in the following topics.

### 6.1. Detection of New Functional Roles of 3′UTRs (3′UTRs as lncRNAs)

Research on RNA is currently a very dynamic field and recent works have highlighted unknown features of 3′UTRs that add some complexity to their function, and to their relationship with other ncRNAs or with regulatory networks. Recent work has suggested that fragments of 3′UTRs could be stabilized in the cytoplasm [228,229] and demonstrated that mature mRNAs were frequently cleaved to generate autonomous 3′UTRs that were polyadenylated and independently regulated by miRNAs (as found for 6068 genes out of 17,393 genes analyzed [230]). This adds complexity to the RNA world and calls for more intense research on the mechanisms generating these autonomous 3′UTRs and their functional interference with the mechanisms ensuring mRNA function.

### 6.2. Improvements in Current 3′UTR Data Curation 

There are generic databases of 3′UTRs sequences that integrate sequence data with motif identification [231] and more specific databases of alternatively polyadenylated 3′UTRs [232,233]. Data on 3′UTR length variation should be integrated with expression data of effectors of 3′UTR dynamics (splice factors, polyadenylation/cleavage factors) extracted from the same microarray or RNA seq. experiments to detect those involved in the generation of 3′UTR variability, especially in disease samples, to characterize the changes in the 3′UTR landscape produced in human disorders [234]. Furthermore, current annotation of genomics and transcriptomics data should be improved to include more data on 3′UTR dynamics, functional motifs and autonomous 3′UTRs. This would require the development of new prediction/recognition algorithms and the analysis and re-annotation of current sequence data.

### 6.3. Generation of New Data on 3′UTR Variants

Most of the data generated to date are subsidiary to standard cloning and sequencing projects that were not specifically designed to cover 3′UTR variability. Although many efforts have been made to facilitate generation of sequencing data, further improvements are required for the effective sequencing of 3′UTR regions, especially in low-input/single-cell samples. This should include the development of new algorithms to facilitate the genome wide profiling and identification of 3′UTRs switching events by alternative polyadenylation/splicing or exonization and their integration with popular genomic browsers [101].

### 6.4. The Landscape of Epitranscriptomic Modifications

In previous sections we have given an overview of structural modifications affecting ribonucleotides in 3′UTRs (A-to-I edition, poly(A) deadenylation, methylation, etc.) that result in translational inactivation or RNA instability. This is to become a hot topic of RNA research that will require new technical developments and advances in sequencing (and consequently in data storage and curation) for its full development. In this sense, it will be interesting to catalogue the epitranscriptomic chemical modifications, to study their effects on RNA function and to characterize the molecular machineries involved. It is clear that, in years to come, RNA research will unveil yet unsuspected mechanisms with an impact on the regulation of gene expression, adding additional layers to the complexity of the RNA world. We anticipate good times on the horizon for research on RNA structure and function.

## Figures and Tables

**Figure 1 biomedicines-09-01560-f001:**
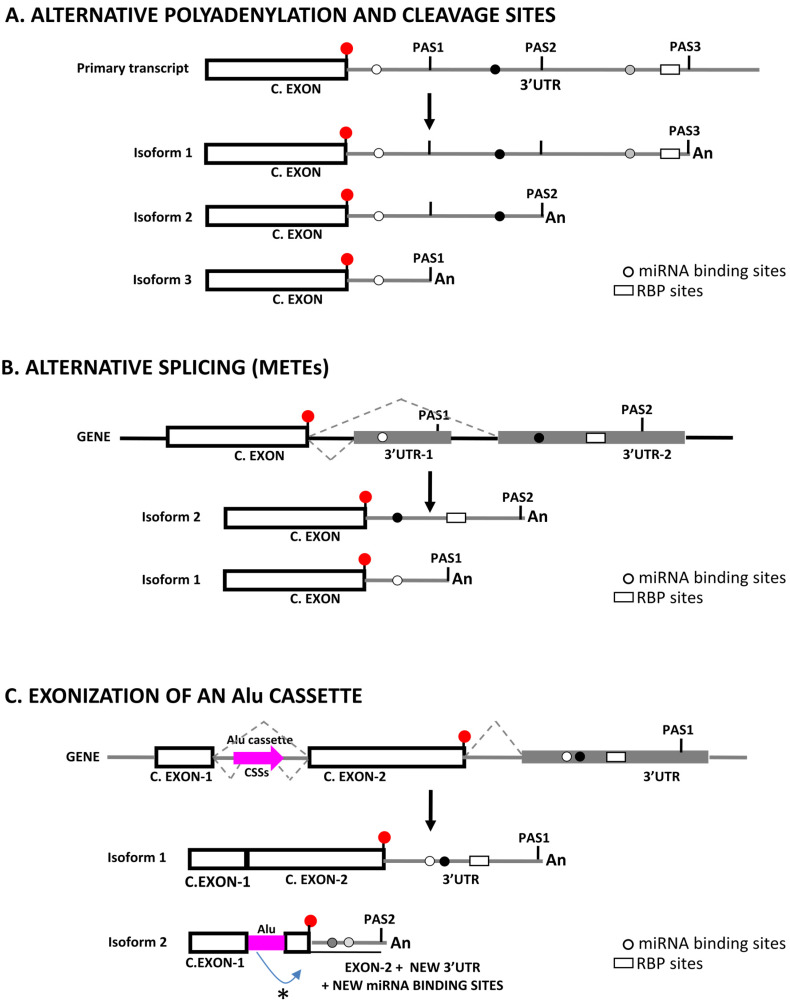
Original drawing showing three mechanisms that generate sequence length variability at 3′UTRs and their potential impact on the regulatory landscape by miRNAs and RBPs. Other mechanisms, or combinations among those displayed are not shown. In all cases C.EXON stands for Coding Exon. (**A**) Use of alternative polyadenylation and cleavage sites. Diagram representing the generation of three different transcript isoforms from a primary transcript by the use of three different polyadenylation and cleavage sites (PAS1, PAS2, PAS3), and how this changes the regulatory landscape. Shown is a terminal coding exon with its stop codon (red dot), as well as the 3′UTRs (gray lines) with the different PAS, miRNA binding sites (gray dots), and a RBP site (white box). [An] stands for the poly-A tail. (**B**) Use of alternative splicing sites: Mutually Exclusive Terminal Exons (METES). Diagram representing the generation of two different transcript isoforms from a single gene by the use of two mutually exclusive 3′UTR terminal exons, and how this changes the regulatory landscape. Shown is a terminal coding exon with its stop codon (red dot), as well as the 3′UTRs (gray boxes/lines) with the different PAS, miRNA binding sites (gray dots), and a RBP site (white box). Dotted gray lines between exons show the two splice events produced. [An] stands for the poly-A tail. (**C**) Exonization of an Alu cassette. Diagram representing the generation of two different transcript isoforms from a single gene by the exonization of an Alu repeated element and how this changes the regulatory landscape. The Alu cassette harbors cryptic splice sites (CSSs) originated by mutation or activated by the unbalance of splicing regulatory elements (see main text for details). In this diagram, exonization of the Alu cassette lead to a change in the ORF and to the activation of a premature stop codon and PAS (shown as *). Shown is the 3′ end of an ideal gene and the two transcripts generated from it. The stop codon is shown as a red dot, the 3′UTRs as gray boxes/lines with the different PAS, miRNA binding sites (gray dots), and a RBP site (white box). Dotted gray lines between exons show the two splice events produced. [An] stands for the poly-A tail.

**Table 1 biomedicines-09-01560-t001:** Mechanisms that generate alternative 3′UTR isoforms and their impact on miRNA binding. Shown are the transcript affected, the mechanism of 3′UTR shortening, the miRNA binding sites lost in the isoform affected, the cells/disease in which the 3′UTR were detected and the reference of the work. See text for more details.

mRNA	Alternative 3′UTR	miRNAs	Cells/Disease	Reference
*Igf1*	APA	LF:miR-29,miR-365	Osteoblastic differentiation	[103]
*CCND1*	APA	LF:over80miRNAs	Mantle cell lymphoma	[104]
*Ki-67*	APA	LF:miR-133-3p,miR-140-3p	Breast cancer	[105]
*FNDC3B*	APA	LF:23miRNAs	Nasopharyngeal carcinoma	[106]
*Tau*	APA	LF:miR-34a	Neuroblastoma cell lines	[107]
*PolH*	APA	LF:miR-619	Lung or bladder cancer	[108]
*ABCB1*	APA	LF:miR-508-5p,miR-145	Leukemia cells	[109]
*ABCC2*	APA	LF:miR-379	Hepatoma cell line	[110]
*COX-2*	APA	n.s.	Colon tumors	[111]
*Hsp70.3*	APA	LF:miR-378	Cardiac ischemic	[112]
*AAMDC*	APA	LF:miR-2428/664a	Adipogenesis	[113]
*Many*	APA *	n.s	Mammalian brain	[114]
*CYP20A1_Alu-LT*	Exonized Alus	Over 140 miRNAs	Primary neurons	[115]
*NR2C1,GTSE1,FHL2,RAD1,FKBP9,CAD,* *SMA4*	Exonized Alus	miR-15a-3p	HeLa cells	[116]
*ADD1andUBE2I*	Exonized Alus	miR-302d-3p	HeLa cells	[116]
*387genes*	Retained intron	n.s.	n.s.	[117]
*CerS1*	Retained intron	miRNA-574-5p	HNSCC	[118]
*BCL11A*	Alternative splicing	ExtraLF:miR-486-3p	Erythroid cells	[119]
*BCL2*	Alternative splicing	LF:miR-204	HCT116 human colon cancercells	[120]
*ADAM12*	Alternative splicing	LF:miR-29andmiR-200	Breast cáncer cells	[121]
*SEMA6Ba*	Alternative splicing	miR-218,miR-19 **	MCF-7cell line	[122]
*CD34*	Alternative splicing	LF:miR-193-5p,miR-125,miR-129-5p,miR-351-5p	Atherosclerosis	[123]

APA: alternative polyadenylation; HNSCC: head/neck squamous carcinoma; LF: long form; n.s.: non-stated; * Very distal polyadenylation sites used, likely encoded by downstream lincRNAs; ** internal sites lost by alternative splicing.

**Table 2 biomedicines-09-01560-t002:** Alu repeat-mediated exonization events causing gene alterations or human disorders. Shown are the transcript affected, the mechanism and effects of Alu exonization and the reference of the work.

GENE	Mechanism or Effects of Alu Exonization	References
*INSL3*	In-frame insertion of an Alu-J *	[201]
*CYP20A1*	Transcript isoform 1 has 23 exonized Alus in its 3′UTR	[115]
*CD58gene*	Alu insertion induces skipping of exon 3 of the *CD58* transcript, originating a frameshifted transcript	[202]
*PKLR*	PK deficiency by activation of a premature stop codon encoded by an exonized Alu Yb9	[203]
*BLOC1S2*	Requires a SINE-MIR for exonization of the Alu element. Exonization activates premature stop codons	[204]
*FVIII*	Mutation in *hnRNPC* binding sites exonizes an Alu Y and truncates F8 protein	[205]
*SMN*	Alu Y element reduces *SMN* mRNA levels (byNMD) in Spinal Muscular Atrophy	[206]
*ATP7Bgene*	Cryptic Alu exon being incorporated into the mature transcript activates a premature stop codon producing a truncated, non-functional protein in Wilson’s disease.	[207]
*GPHA*	In-frame Alu-J element exonization increases length of the N-terminus and enhances the bioactivity of HCG protein	[208]
*COL4A5*	Downstream deletion induces exonization of an Alu-Y element encoding a stop codon, resulting in a truncated protein in Alport Syndrome	[209]
*NSUN2*	AluY element reduces *NSUN2* mRNA levels and results in Dubowitz syndrome	[210]
*REL*	Increased transactivation activity by two fold	[211]
*CETP*	In-frame insertion of an Alu element *	[212]
*NARF*	RNA editing creates a functional AG3′ splice site, and eliminates a premature stop-codon in the Alu exon element.	[213]
*Survivin2*	Alu exon being incorporated into the mature transcript activates a premature stop codon, making it susceptible to NMD	[214]
*GMRalpha*	In-frame inclusion of an Alu-element increases ORF by 34 aminoacids in the extracellular domain of GMRalpha preferentially targeted by ectodomain proteases	[215]
*ACE*	n.d.	[216]
*Bcl-rambobeta*	Truncated protein by activation of a premature stop codon in exonized AluY	[217]
*DMD*	Dystrophin deficiency (X-linked dilated cardiomyopathy) by activation of a premature stop codon encoded by the exonized Alu	[218]
*GUSB*	Mild mucopolysaccharidosistype VII (MPSVII) by premature termination of β-glucuronidase translation	[219]
*RED1/ADAR2*	Exonization of an in-frame Alu-J cassette in the deaminase domain reduces catalytic activity	[186]
*DRADA2*	In-frame insertion of an Alu element *	[220]
*COL4A3*	COL4A3 deficiency by activation of a premature stop codon encoded by the exonized Alu in Alport Syndrome	[221]
*BGP*	In-frame insertion of an Alu element *	[222]
*CHRNA3*	α-3AChR deficiency by activation of a premature stop codon in the exonized Alu	[223]
*Nramp*	NRAMP deficiency by activation of a premature stop codon encoded by the exonized Alu	[224]
*OAT*	OAT deficiency by activation of a premature stop codon encoded by the exonized Alu	[225]
*DAF*	Generation of alternative C-terminal domains and 3′UTRs in alternative DAF isoforms	[226]
*Complement-C5*	In-frame insertion of an Alu element *	[227]

This is a non-systematic recompilation of exonized Alus in 3′UTRs and includes only elements exonized in mRNAs but not in ncRNAs and with an associated phenol type. See text for more details. Abbreviations: n.d., non-determined. (*) in-frame insertion creates longer transcripts.

## Data Availability

Not applicable.

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
