# Peer review of "Dynamic Variations of 3′UTR Length Reprogram the mRNA Regulatory Landscape"

_biomedicines, 2021, doi:10.3390/biomedicines9111560_

Round 1
Reviewer 1 Report
Navarro et al. review the literature on 3’UTR length control, their molecular impact, and role in human diseases. Overall this is good overview of the field, but I have several comments for improvement.
- Only mRNA stability is discussed for 3’UTR sequence implications in part 2. However, the authors do mention “spatiotemporal translation” control in the abstract. Discussion of mRNA localization and translation activation/repression by motifs and RBPs would offer a more complete picture.
- Line 39, should this be PAP instead of PNPT1?
- Figure 1, multiple miRNA binding sites with different color may be more distracting to the authors’ points than helpful.
Author Response
Dear reviewer, thanks for your helpful comments that have improved the manuscript. Those are my answers:
1.- With regard of the "spatiotemporal translation” control, we have added a full new section (section 3) on the topic.
2.- PAP vs. PNPT1. You were right. This has been corrected
3.- The drawing has been corrected and the colours eliminated. We hope that these changes make the figure easier to understand
Reviewer 2 Report
This review provides an update of the current knowledge on the mechanisms of regulation of the size of the 3'UTR regions of mRNAs. This review is well written and reports the main part of the data known to date.
It might be interesting in the proposed perspectives to introduce the study of epitranscriptomic modifications that can act on RNA splicing and stability. The location of these modifications and their frequency could also influence the size of the 3'UTR regions.
Author Response
Dear reviewer, thanks for your comments that have improved the manuscript.
We have added a new perspective, as suggested, and in the new section 3 (spatio/temporal control of mRNA function) we have make a brief citation of the deadenytaling/methylating machineries. We think that this is such an important topic that it would deserve a full review in itself.